# Stability of Intracellular Protein Concentration under Extreme Osmotic Challenge

**DOI:** 10.3390/cells10123532

**Published:** 2021-12-14

**Authors:** Jordan E. Hollembeak, Michael A. Model

**Affiliations:** Department of Biological Sciences, Kent State University, Kent, OH 44023, USA; jhollemb@kent.edu

**Keywords:** cell volume regulation, macromolecular crowding, transport-of-intensity equation, transmission-through-dye, intracellular water, vacuolization

## Abstract

Cell volume (CV) regulation is typically studied in short-term experiments to avoid complications resulting from cell growth and division. By combining quantitative phase imaging (by transport-of-intensity equation) with CV measurements (by the exclusion of an external absorbing dye), we were able to monitor the intracellular protein concentration (PC) in HeLa and 3T3 cells for up to 48 h. Long-term PC remained stable in solutions with osmolarities ranging from one-third to almost twice the normal. When cells were subjected to extreme hypoosmolarity (one-quarter of normal), their PC did not decrease as one might expect, but increased; a similar dehydration response was observed at high concentrations of ionophore gramicidin. Highly dilute media, or even moderately dilute in the presence of cytochalasin, caused segregation of water into large protein-free vacuoles, while the surrounding cytoplasm remained at normal density. These results suggest that: (1) dehydration is a standard cellular response to severe stress; (2) the cytoplasm resists prolonged dilution. In an attempt to investigate the mechanism behind the homeostasis of PC, we tested the inhibitors of the protein kinase complex mTOR and the volume-regulated anion channels (VRAC). The initial results did not fully elucidate whether these elements are directly involved in PC maintenance.

## 1. Introduction

Short-term cell volume (CV) regulation has been studied extensively since the 1960s. Since water is the most abundant intracellular substance, CV is primarily determined by the intake or expulsion of intracellular water. The amount of water is, in turn, dictated by the requirement to maintain isosmolarity with the environment. The typical experiment involves subjecting cells to an abrupt change in external osmolarity. Because most cell membranes are highly permeable to water, the cell’s internal osmolarity quickly readjusts either by accumulating water in a hypoosmotic solution or by releasing water in a hyperosmotic solution. As a result, cell volume changes. It has been found, however, that cells often partially or fully restore their initial volume. Thus, swelling in a hypoosmotic solution is followed by shrinkage, which is termed the regulatory volume decrease (RVD); conversely, shrinkage in a hyperosmotic solution can be followed by the regulatory volume increase (RVI). RVI and RVD are mediated, respectively, by the accumulation or release of ions or other osmotically active substances [1]. The latter processes do not change the osmolarity of the cytoplasm (which remains equal to the new external osmolarity) but allow for a second phase of water redistribution and restoration of CV.

RVD and RVI in growing cell cultures are usually monitored only for a brief period, often as short as several minutes, which helps avoid the ambiguity resulting from cell growth and division. In some cases, the regulatory volume changes are absent during the observation time, which leaves the question open as to what happens over a longer period.

One way to address this question is to compare the average cell volumes of large populations. However, if one is interested in the volume behavior of individual cells in a growing culture, an arguably more informative approach is to replace CV with cell water content or with the concentration of intracellular dry matter (which we can call “protein” for brevity). These two quantities are interrelated in a simple way: protein concentration (PC) can be expressed through the relative volume occupied by water [2]:$$\mathrm{PC}\left(\frac{g}{\mathrm{ml}}\right)=\frac{M_{p}}{\mathrm{CV}}\approx 1.4\left(1-\frac{V_{\mathrm{water}}}{\mathrm{CV}}\right)$$

During rapid responses, PC and CV are tightly linked and carry almost identical information because protein mass (M_p)_ does not have time to change; however, during prolonged observations, PC becomes a much more meaningful parameter. Focusing on PC makes all the more sense as it is believed to have a major and direct impact on numerous cellular processes. At PC levels as high as those observed in living cells (the condition known as “macromolecular crowding”), the effective concentration, or thermodynamic activity of proteins becomes a steep function of PC, so that even a minor shrinkage or swelling is expected to have disproportionately strong effects [3]. Namely, an increase in PC (which corresponds to relative dehydration) favors more compact protein shapes and the formation of protein complexes. These effects have been abundantly demonstrated in vitro [4,5] but the understanding of the exact role of macromolecular crowding in living cells is still at a very early stage [6].

In this work, we report PC measurements in cells cultured for 24–48 h in hyperosmotic or hypoosmotic media. A similar study has been done on E. coli using a fluorescent crowding sensor [7] but to our knowledge not on mammalian cells. We found that PC remains surprisingly conserved within a wide range of external osmolarities, both for HeLa and 3T3 fibroblasts. Next, we asked what happens when the osmotic stress becomes unmanageable. Finally, we describe some experiments aimed at an understanding of the molecular mechanisms responsible for PC stability.

## 2. Reagents and Methods

### 2.1. Cell Culture

HeLa or 3T3 fibroblasts (ATCC, Manassas, VA, USA) were grown in Dulbecco’s Modified Eagle’s Medium (DMEM) either from Lonza BioWhittaker (Walkersville, MD, USA) or from Corning (Corning, NY, USA) supplemented with 10% fetal bovine serum (FBS, Fisher Scientific, Waltham, MA, USA) and antibiotics. To prepare hypoosmotic media, DMEM was diluted with water, or in some cases, with 10% FBS. Hyperosmotic media were prepared by adding sucrose to DMEM.

Osmolarity was not measured directly but estimated based on the manufacturer’s data for DMEM (340 mosm/kg for either source) and assuming ideal osmotic behavior. When 10% FBS, rather than water, was used to prepare dilutions, the numbers for osmolarity were adjusted accordingly.

The following reagents were used: gramicidin, fluorescein sodium, Hoechst 33258, DCPIB, and DiOC_18_(3) (3,3′-dioctadecyloxacarbocyanine perchlorate; DiO), all from Sigma-Aldrich (St. Louis, MO, USA); torin 1 and rapamycin from ApexBio (Houston, TX, USA); 4,4′-diisothiocyano-2,2′-stilbenedisulfonic acid (DIDS), amiloride, and cytochalasin D from Cayman Chemical (Ann Arbor, MI, USA); Acid Blue 9 (TCI America, Portland, OR, USA); NucView 488 (Biotium, Fremont, CA, USA); dextran Alexa 488 (Invitrogen, Eugene, OR, USA); tetramethylrhodamine ethyl ester (TMRE; Biotium, Hayward, CA, USA), and calcein AM ester (AnaSpec, Fremont, CA, USA).

### 2.2. Cell Volume and Protein Concentration Measurements

CV and cell dry mass were determined as described in detail previously [8,9,10]. Cells were grown on loose coverslips, which before observation were mounted on slides in their incubation media containing additionally ~7 mg/mL Acid Blue 9. They were imaged on an IX81 microscope (Olympus, Center Valley, PA, USA), usually with a 20×/0.7 objective, either through a 630/10 filter for volume measurements by transmission-through-dye (TTD) or through a 485/10 filter for dry mass measurements by the method based on the transport-of-intensity equation (TIE) [11,12]. The intensity I_ij_ in TTD images was converted to cell depth units h_ij_ and cell volume as
$$h_{\mathrm{ij}}=\frac{\ln\left(I_{\mathrm{ij}}-I_{\mathrm{dark}}\right)}{\alpha }$$
$$\mathrm{CV}=\underset{A}{\sum }(h_{\mathrm{ij}}-h_{\mathrm{bkg}})=A\left(\bar{h}-h_{\mathrm{bkg}}\right)$$

In this equation, α is the absorption of the dye solution determined by lens immersion [13], A is the cell area, and the bar over h indicates the average; the subscripts ij, bkg, and dark refer to the cell, the background near the cell, and the intensity obtained with the shutter closed, respectively. Image processing was performed using ImageJ (https://imagej.nih.gov/ij/, version 1.53k). TTD images are shown on a logarithmic intensity scale, where cell thickness is directly proportional to brightness.

The TIE technique measures the cell refractive index, which is related to protein concentration; it utilizes variations in the cell appearance at different focal distances when viewed under a bright-field microscope [14]. Different TIE algorithms have been devised; our analysis was based on two bright-field images: the first one was focused on the plane of the coverslip, and for the second one, the objective was brought towards the sample by 5 μm. The computation was performed using MatLab (The MathWorks, Inc.; Natick, MA, USA) using the code of Gorthi and Schonbrun [15]; the details of the TIE measurements are described in the Appendix A, Section 2. The ratio of the cell-integrated TIE value to the corresponding volume gives PC, with a calibration factor that may have to be determined empirically. Examples of TTD and TIE images are given in Figure 1; the details of the TIE calibration are described in the Appendix A.

### 2.3. Fluorescence Microscopy

Fluorescence imaging of apoptotic markers was performed on the same IX81 wide-field microscope under illumination from a Hg lamp; the emission was collected through appropriate fluorescence filters, and the images were captured by a cooled CCD camera (Cooke, Romulus, MI, USA); confocal fluorescence images were collected on laser scanning confocal systems Fluoview 1000 or Fluoview 3000 (Olympus, Center Valley, PA, USA).

### 2.4. Estimation of Cell Survival

Cells were grown in multi-well culture plates. Regions were marked on the dish bottom, and cells in those regions were counted before and after treatment.

## 3. Results

### 3.1. PC Homeostasis

Hyperosmotic media were prepared by the addition of sucrose and tested for 24 h. Hypoosmotic conditions were tested for HeLa cells for 24–48 h in DMEM diluted either with water or 10% FBS; 3T3 fibroblasts were only tested in water-diluted DMEM for 24 h. HeLa retained healthy morphology and proliferated at a normal rate in 50% DMEM; at greater dilutions or high hyperosmolarity, cells eventually began to detach. Since we were interested primarily in the PC, the analysis was limited to the remaining cells.

Like most other eukaryotic cells, HeLa and 3T3 fibroblasts have PC ≈ 0.2 g/mL in standard media [6]. It is known that adherent HeLa show vigorous RVD in hypoosmotic media [16,17] but lack RVI in hyperosmotic [9,18]; their long-term behavior has not been characterized, except for some limited data previously obtained in our lab [9]. Figure 2 shows the relative changes in PC following prolonged incubation in media with different osmolarities. We found that PC remained conserved in the entire range of tested osmolarities except for 25% DMEM for HeLa, when it began to increase. The use of 10% FBS to prepare 25% media slightly mitigated the increase in PC. In one experiment, the incubation time was extended from 24 to 48 h; this had no effect on HeLa in 50% DMEM, but an increase in PC became noticeable in 33% DMEM. No change in PC was observed in most extreme hyperosmotic solutions (up to 300 mM sucrose). Examples of TTD images showing cell morphology after a 24-h incubation at different conditions are given in Figure 3. 3T3 fibroblasts behaved similarly, except that the range of tolerated osmolarities was narrower, and the detachment preceded a possible PC elevation in highly dilute media.

Thus, despite the initial water accumulation caused by the hypoosmotic environment, cells recover and maintain their PC over a long period; paradoxically, an extreme hypoosmolarity causes a loss of water and an increase in PC (in the cells remaining on the coverslip). Hyperosmolarity does not result in major changes in PC.

A qualitatively similar response was elicited by cationic ionophore gramicidin [19]. Gramicidin causes immediate intracellular sodium accumulation and osmotic swelling; however, within 24 h, swelling gives way to shrinkage, whose extent depends on gramicidin concentration (Table 1). Cell morphology remains little changed even at the highest of the tested gramicidin concentrations (Figure 3D).

### 3.2. Response of HeLa to Extreme Osmolarities

It follows from our results that as long as cells are healthy and viable, their PC remains constant; however, when they come under excessive hypoosmotic stress, they undergo dehydration and an increase in PC. Since dehydration is a typical apoptotic feature [1], we tested if incubation of HeLa cells under highly hypoosmotic (33% DMEM) or hyperosmotic (100% DMEM + 200 mM sucrose) conditions results in the activation of apoptotic processes: caspase-3 activation and condensation of chromatin. No caspase-3 staining was noticed in the cells remaining on the substratum (Figure 4A,B), and only occasional detached cells that came into the field of view (Figure 4C) showed NucView staining.

At the same time, prolonged incubation in strongly hypoosmotic media (such as 25% and sometimes 33% DMEM) resulted in the development of prominent vacuoles in HeLa cells (only cells without large vacuoles were analyzed for Figure 2). Similar vacuoles were observed in moderately hypoosmotic media (50% DMEM) in the presence of the actin depolymerizing drug cytochalasin D (Figure 5). TIE revealed that these vacuoles contain hardly any protein, while the surrounding cytoplasm remains at normal density.

This leads to the question of the origin of these vacuoles. The best-characterized vacuoles in mammalian cells result from aberrant pinocytosis, a process known as methuosis [20,21]. Methuotic vacuoles would be expected to accumulate fluorescent dextran, which is a classical marker of endocytosis. Thus, we carried out the incubation in the presence of 10 μg/mL dextran-Alexa 488 and imaged cells in the same media under a confocal microscope. Some spotty cytoplasmic staining was indeed observed, though not coinciding with the vacuoles: the large vacuoles remained dark (Figure 6A,B). An inhibitor of pinocytosis amiloride, when applied in hypoosmotic media, caused rapid apoptosis (Appendix A).

We next asked if these vacuoles are surrounded by a lipid membrane or represent a phase transition, as in necrotic blebs [22]. One way to approach this question would be to stain them with a fluorescent marker and test their mobility by fluorescence recovery after photobleaching, as was done in the aforementioned work. However, calcein AM did not load into the vacuoles (Figure 6C); neither did sodium fluorescein or CFSE (not shown). We tried to detect the presence of a membrane by direct lipid staining with DiO. No clear evidence of a membrane was found; where the vacuoles were surrounded by a continuous fluorescent layer, it was probably the nuclear membrane or small cytosolic vesicles (Figure 6D,E). On the other hand, the exclusion of calcein still suggests the existence of a membrane.

### 3.3. The Effect of Inhibitors

Several inhibitors were tested for their effects in dilute (33–50%) solutions: mTOR inhibitors rapamycin and torin 1, the blockers of volume-regulated anion channels (VRAC), DIDS and DCPIB, a potassium channel blocker TEA, and a sodium channel blocker amiloride. No consistent changes in PC were observed with the first five inhibitors (Table 2). However, in two experiments with torin 1, the average PC dropped to below 0.6; in another experiment, it increased to 1.22. Likewise, in the presence of DIDS, the values ranged from 0.73 to 1.26. A sodium channel inhibitor amiloride was tested as well. It caused an increase in PC and rapid apoptosis in 50% DMEM (Appendix A).

## 4. Discussion

The presented study is perhaps the first characterization of the PC behavior in mammalian cells subjected to anisosmotic stress. Numerous researchers have previously investigated biological responses caused by prolonged anisosmolarity; it has been determined, for example, that cells become smaller in hypoosmotic and larger in hyperosmotic media [23,24], but those studies did not extend to PC. The relative neglect of PC in favor of CV may have been due to the difficulty of measuring it in live cells. However, the availability of TIE/TTD opens new experimental possibilities. The technique is simple enough to be used in most labs and requires only a transmission light microscope with precision vertical travel and two optical filters. Unlike fluorescence-based techniques for assessing macromolecular crowding [25,26], TIE/TTD directly reports the macromolecular density averaged over cell thickness.

We know from in vitro data that PC affects biochemical reactions through alterations in molecular conformation, aggregation, and reaction rates [3], but data on living cells have been scarce. PC values around 0.2 g/mL have been reported for most diverse eukaryotic cell types [6] and a similar consistency has been proposed for microorganisms [27]. Here we have shown that, in addition to that, cells keep stable cytoplasmic PC in the face of extreme osmotic stress; this fact can be interpreted as yet another indication that PC ≈ 0.2 g/mL is important enough to be worth maintaining.

However, the only way to identify the role of a biological factor is to alter it. Persistent shrinkage is a well-known apoptotic trigger [28], and we have recently shown that high PC can act as a direct apoptotic stimulus [8]. It seems that various modes of cell deterioration, such as apoptosis caused by other factors [1], cell senescence [29,30], or unspecified damage resulting from very low osmolarity (Figure 2) or gramicidin (Table 1) eventually result in dehydration and an increase in PC; thus, dehydration seems to be a common cellular response to stress. This fact is remarkable because the natural cellular reaction to ATP depletion or to membrane failure is an accumulation of water rather than its expulsion [31]; therefore, a loss of water from stressed cells must be a purposeful response to adverse conditions. The dehydration response has been particularly thoroughly studied in apoptosis, where it is termed apoptotic volume decrease, or AVD [1,32,33]. Like RVD, AVD depends on the efflux of osmolytes, such as potassium and chloride; most researchers believe that K^+^ channels become directly activated as a result of apoptotic signaling [34,35], though this view is not universally shared [36,37]. The presumed benefit of AVD is the facilitation of phagocytosis [38] or prevention of swelling and rupture that might otherwise develop in damaged cells [39].

AVD is often observed early in apoptosis, before the onset of other detectable changes [40]. Dehydration observed in our experiments clearly preceded the classical signs of apoptosis—caspase activation and condensation of chromatin (if there were any), and in the case of gramicidin, cell morphology was not even visibly affected. Thus, the observed increase in PC could have been an early AVD, although we did not attempt to follow the cells’ fate after their detachment and to determine the mode of their death.

Keeping cells in exceedingly dilute solutions (25% DMEM) produces another interesting effect; instead of uniformly spreading throughout the cell, water forms separate pools, while the intervening cytosol remains at nearly normal density. Similar vacuoles were observed in moderately dilute, 50% DMEM when actin depolymerizing drug cytochalasin was added; apparently, the cytoskeleton helps cells resist water influx [37]. Since the standard marker of pinocytosis, fluorescent dextran, does not accumulate there, we concluded that these vacuoles do not originate from pinocytic vesicles. So far, we have found no convincing evidence of a membrane surrounding the vacuoles, though the exclusion of hydrophilic calcein suggests so. Regardless of the nature of the vacuoles, it seems that cells resist prolonged absorption of extra water into the cytosol and either expel it outside or segregate it into separate compartments.

We have previously observed a similar situation in severely compromised cells at late stages of free-radical-induced damage when necrotic blebs gradually lose proteins [22]. That situation was apparently caused by a decrease in protein solubility and their extensive aggregation. Whether we have here a case of inner necrotic blebbing or some other process, can only be determined by additional studies.

The mechanism of long-term PC homeostasis is equally intriguing and not necessarily identical to rapid RVI or RVD. Here we focused mostly on a hypoosmotic environment. Prolonged exposure to external hypoosmolarity as low as one-third of normal or to gramicidin-mediated accumulation of sodium did not produce dilute cells. Even though many cells started dying at osmolarities below half of normal, the remaining cells kept their normal density or even overcompensated, expelling additional amounts of water. We have therefore tested several inhibitors that could be expected to oppose PC recovery. Our choice of inhibitors (Table 2) was guided by the following considerations. PC, by definition, is the ratio of protein to volume, and a simple model of osmotic balance leads to the equation [6]:$$\mathrm{PC}=\frac{N_{p}\mu_{p}\left(E-\left[I\right]\right)}{N_{o}}$$

Here, N_p_ is the number of protein molecules, N_o_ is the number of small osmolytes (such as amino acids), μ_p_ is the average molecular mass of proteins, E is the external osmolarity, and [I] is the cumulative intracellular concentration of inorganic ions. The difference from CV is in the factor N_p_μ_p_, which points to an additional means of PC regulation, compared to CV regulation. For example, stimulation of protein degradation can be expected to lower PC. This can act through one of the two mechanisms (or their combination): a decrease in organic mass, if the products of proteolysis are removed from the cell, or an increase in osmolytes and water, if they are retained (note that TIE would not tell the difference between proteins and amino acids; however, under normal conditions, the mass of intracellular proteins is much greater). The best-characterized switch between anabolism and catabolism is the protein kinase complex mTOR; its inhibition stimulates proteolysis [41], and we expected that mTOR inhibitors, rapamycin or Torin 1, might promote water accumulation in hypoosmotic solutions.

The other possible target would be ion (potassium and chloride) and/or organic osmolyte channels, such as the much-discussed volume-regulated anion channels (VRAC). Their inhibition by DIDS and DCPIB [42] could in principle cause the retention of osmolytes and water. However, both channel blockers and mTOR inhibitors failed to reliably reduce protein density, though in some cases the effect was observed. In two experiments where mTOR inhibitors were added to 33% DMEM, we indeed observed a 40% decrease in PC. In two other experiments, an increase in PC occurred instead, which was quite significant in some cells. In yet other cases, there was little change. We speculate that a lack of good reproducibility may have resulted from a tug-of-war between two opposing processes with uncertain outcome: inhibition of mTOR may prevent the restoration of PC in dilute media, but if cell damage occurs in the process, cells respond by water expulsion and an increase in PC.

## 5. Conclusions

HeLa and 3T3 fibroblasts recover their normal intracellular protein concentration (and hence water content) and maintain it during prolonged incubation in strongly hyperosmotic or hypoosmotic solutions.

When subject to an extreme hypoosmotic stress (~85 mosm/L for HeLa) or high concentrations of the Na^+^/K^+^ ionophore gramicidin, cells eventually increase their protein density. We hypothesize that an increase in PC and water expulsion may be a general cell response to severe stress.

At very low external osmolarity or when weakened by dissolution of the cytoskeleton, cells segregate water into protein-free vacuoles rather than absorb it into the cytosol.

## Figures and Tables

**Figure 1 cells-10-03532-f001:**
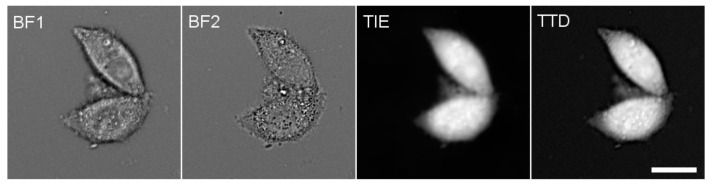
The successive steps of PC measurement. BF1 and BF2 are two mutually defocused bright-field images of HeLa cells taken through a 485 nm filter and used to compute a TIE image, which represents cell dry mass. Independently, a TTD image representing cell thickness was collected through a 630 nm filter. The total cell intensities were computed on both TIE and TTD images, and their ratio was multiplied by an empirical calibration factor 0.734 to obtain PC (see Appendix A). In most experiments reported in this work, only relative changes in PC are given. Scale bar, 20 μm.

**Figure 2 cells-10-03532-f002:**
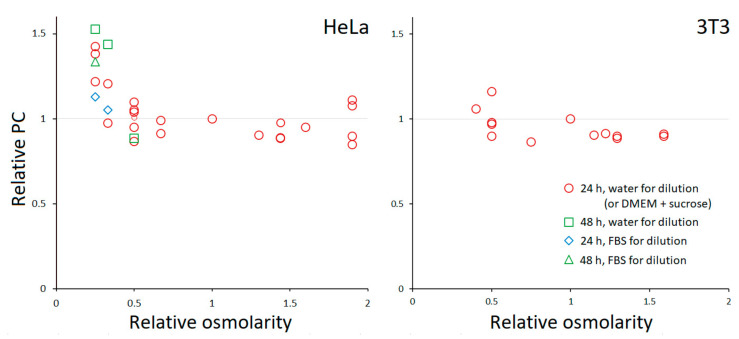
The dependence of PC on the osmolarity of the medium. Each point represents the average for a separate experiment, in which usually 5–15 cells or cell groups were analyzed. PC was normalized to that in the control (100% DMEM with 10% FBS). The coefficient of variation for a single experiment was typically between 0.05 and 0.1.

**Figure 3 cells-10-03532-f003:**
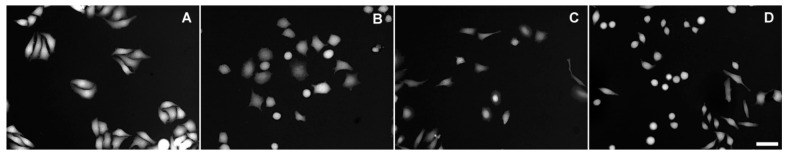
TTD images of HeLa cells exposed to various conditions for 24 h. (**A**) Untreated; (**B**) 100% DMEM + 0.3 M sucrose; (**C**) 33% DMEM; (**D**) 100% DMEM + 2.5 μM gramicidin. Scale bar, 50 μm.

**Figure 4 cells-10-03532-f004:**
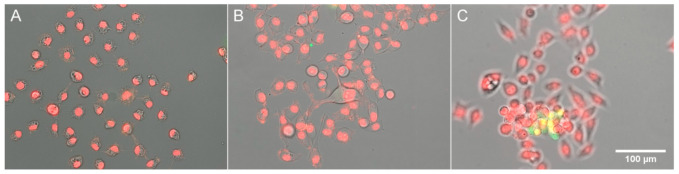
Staining for apoptosis in HeLa cells. Hoechst staining of DNA is shown in red for better visibility; caspase staining is shown in green. (**A**) 33% DMEM; (**B**,**C**) DMEM + 0.2 M sucrose.

**Figure 5 cells-10-03532-f005:**
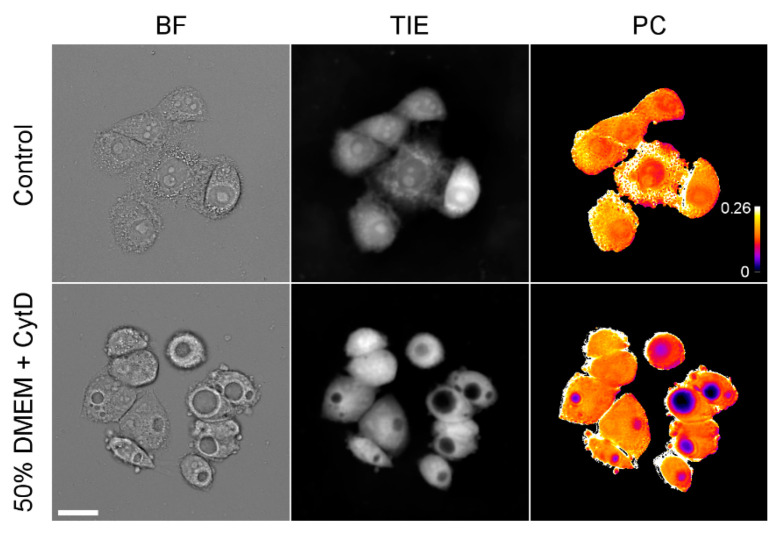
Images of HeLa cells incubated for 24 h in 50% DMEM containing 0.4 μM cytochalasin D. In-focus bright-field images, TIE images, and semi-quantitative PC maps are shown. The calibration bar shows an approximate (because pixel-by-pixel division of images was done only approximately) protein concentration in g/mL. Scale bar, 25 μm.

**Figure 6 cells-10-03532-f006:**
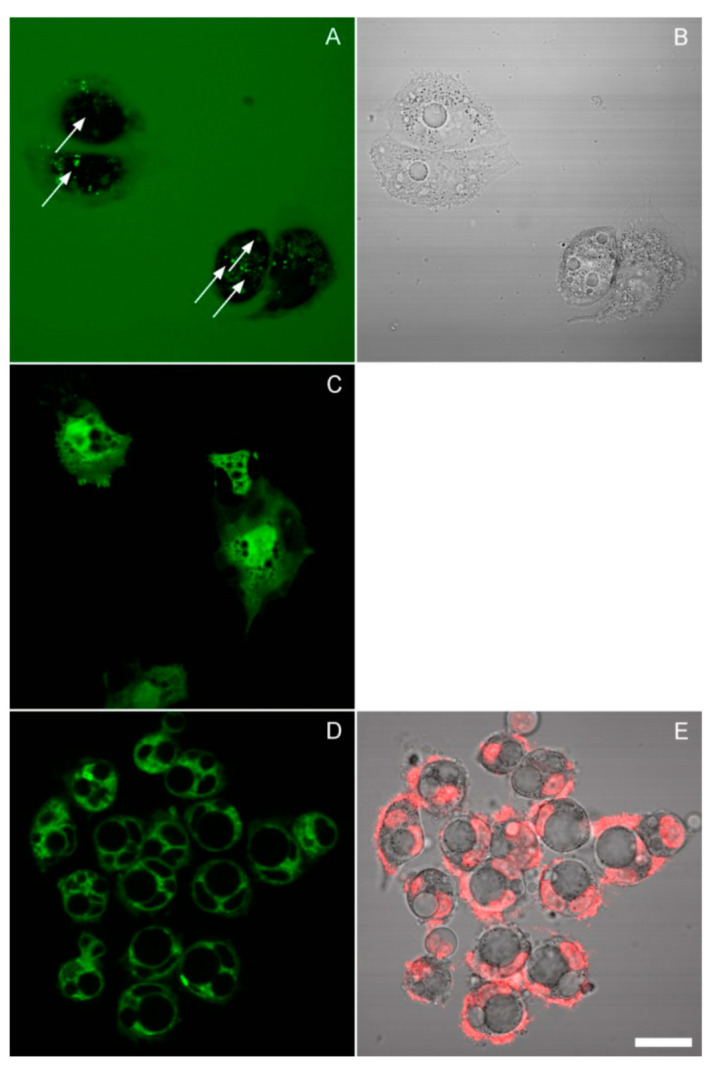
Images of HeLa cells collected by confocal scanning using a 60×/1.2 water-immersion objective. (**A**,**B**) The cells were incubated for 24 h in 50% DMEM/cytochalasin D in the presence of fluorescent dextran. The arrows in the fluorescence image (which represents a maximum intensity projection of several planes) point to the centers of vacuoles identified on the corresponding transmission bright-field image. Bright spots of internalized dextran lie outside the vacuoles. (**C**) Calcein staining of cells preincubated for 24 h in 25% DMEM. (**D**,**E**) Cells were preincubated for 24 h in 50% DMEM/cytochalasin and stained with DiO; image E shows a superposition of Hoechst and transmission. The thin lines visible around some of the vacuoles also border the nuclei, and since a similar DiO pattern was observed in untreated cells (not shown), they probably represent the nuclear membranes. Scale bar, 25 μm.

**Table 1 cells-10-03532-t001:** The effect of gramicidin on the relative changes in PC of HeLa cells.

Gramicidin Concentration (μM)	PC (Relative)		
	25 min	1 h	24 h
0.1	0.93	0.945	1.085 ± 0.054 (3)
0.5	0.87	0.99	1.121 ± 0.122 (3)
2.5	0.71	0.60	1.234 ± 0.135 (3)

**Table 2 cells-10-03532-t002:** The effect of inhibitors on PC in dilute solutions. Combined results of several experiments are shown. The effects of these inhibitors were tested in 33–50% DMEM; those results were combined as well.

Inhibitor	PC (Relative) Mean ± SD (n)
torin 1 (20–100 nM) or rapamycin (100 nM)	0.93 + 0.24 (6)
DCPIB (10 μM) or DIDS (0.2–1 mM)	0.99 + 0.17 (5)
TEA (10 mM)	0.95 (1)

## Data Availability

Original data will be provided on request.

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
