# Peer review of "Stability of Intracellular Protein Concentration under Extreme Osmotic Challenge"

_cells, 2021, doi:10.3390/cells10123532_

Round 1
Reviewer 1 Report
​This is for the most part a very useful, thoughtful and well executed paper, but there are some confusing parts.
General points. The author's use of the term 'dehydration' caused me some confusion. At first I thought that it meant that the cytoplasm was hypertonic relative to outside saline but that does not seem to be the case. It seems that the authors are assuming that after the change of the extracellular osmolarity that the intracellular osmolarity changes to match that of the extracellular solution by osmosis. This is a very reasonable assumption. If this is what they think happens they should say it. If not they need to clarify what they are saying.
What seems to be happening in the case of a hypotonic saline is that the cells reduce the osmolarity of the cytoplasm by allowing the efflux of ions (K+,Na+,Cl- etc.) and osmolytes ( taurine, amino acids etc. ) & the higher PC may simply be the cell attempting to increase the ionic strength of the cytoplasm. It seems to be true that the protein is becoming relatively dehydrated, but the whole cell is more hydrated.
The authors should at some point note that proteins only contribute about 5 mOsm to the cytoplasmic osmolarity ( see e.g. https://onlinelibrary.wiley.com/doi/full/10.1002/bies.201300066)
The equation on line 305 really does not help one understand what is happening - I'd suggest dropping it.
Specific points
L93 - reference needed
Fig 2 - I suggest using different colors or shades to designate different experimental conditions on the graph
L222 presumably you mean liquid-liquid phase separation? if so add a reference
Reviewer 2 Report
Hollembeak and Model describe their analysis of the protein concentration in cells that are long-term exposed to hyper- and hypoosmotic stress. They find that the protein concentration does not change, while more substantial hypoosmotic stress increases the protein concentration. These findings are intuitive, apart from the increased protein concentration at the lowest osmolarities.
Although a deeper mechanistic insight has not been identified, these findings are an important building block for future studies.
The core technology of the paper is TIE that is used to determine protein concentration. The TIE depends on the refractive index. This also means that salts affect the readouts, as well as metabolites (and amino acids, as the authors refer to in the discussion). Can the authors estimate how much salts (e.g., KCl) and small molecules would perturb the readouts at their physiological concentrations? How much do they perturb the readouts reported in Figure 2, and in which direction?
In addition, one may speculate that biomacromolecules form metabolons, phase separations, aggregates and change their organization under some of the conditions described in the paper. How would such an effective change in particle size (and number density, for that matter) influence the measured refractive index?
Figure 2 contains 5-15 cells for each data point: Would the authors mind adding the error bars?
Tables 1 and 2 need a bit more information. It is understandable, but instead of µM, it should read: Concentration Gramicidin (µM). Also, the heading for the three other columns should read PC (relative).
The authors describe several mechanisms that do not cause the observed phenomena, or at least not robustly. It would be great if the authors could speculate on a mechanism of what could cause the higher protein concentration at low external osmolarity in the discussion section. In addition, what could be the reason for the cells to pursue dehydration at low osmolarities?
Round 2
Reviewer 2 Report
No further comments.